# Improving prediction of response to neoadjuvant treatment in patients with breast cancer by combining liquid biopsies with multiparametric MRI: protocol of the LIMA study – a multicentre prospective observational cohort study

Liselore M Janssen [1], Britt B M Suelmann,[2] Sjoerd G Elias,[3] Markus H A Janse,[1] Paul J van Diest,[4] Elsken van der Wall,[2] Kenneth G A Gilhuijs[1]

For numbered affiliations see end of article.

**Correspondence to**
Dr Kenneth G A Gilhuijs;
K.G.A.Gilhuijs@umcutrecht.nl

## ABSTRACT

**Introduction** The response to neoadjuvant chemotherapy (NAC) in breast cancer has important prognostic implications. Dynamic prediction of tumour regression by NAC may allow for adaption of the treatment plan before completion, or even before the start of treatment. Such predictions may help prevent overtreatment and related toxicity and correct for undertreatment with ineffective regimens. Current imaging methods are not able to fully predict the efficacy of NAC. To successfully improve response prediction, tumour biology and heterogeneity as well as treatment-induced changes have to be considered. In the LIMA study, multiparametric MRI will be combined with liquid biopsies. In addition to conventional clinical and pathological information, these methods may give complementary information at multiple time points during treatment.

**Aim** To combine multiparametric MRI and liquid biopsies in patients with breast cancer to predict residual cancer burden (RCB) after NAC, in adjunct to standard clinico-pathological information. Predictions will be made before the start of NAC, approximately halfway during treatment and after completion of NAC.

**Methods** In this multicentre prospective observational study we aim to enrol 100 patients. Multiparametric MRI will be performed prior to NAC, approximately halfway and after completion of NAC. Liquid biopsies will be obtained immediately prior to every cycle of chemotherapy and after completion of NAC. The primary endpoint is RCB in the surgical resection specimen following NAC. Collected data will primarily be analysed using multivariable techniques such as penalised regression techniques.

**Ethics and dissemination** Medical Research Ethics Committee Utrecht has approved this study (NL67308.041.19). Informed consent will be obtained from each participant. All data are anonymised before publication. The findings of this study will be submitted to international peer-reviewed journals.

**Trial registration number** NCT04223492.

## STRENGTHS AND LIMITATIONS OF THIS STUDY

⇒ The LIMA trial aims to improve prediction of response to neoadjuvant treatment in patients with breast cancer by combining liquid biopsies with multiparametric MRI.
⇒ LIMA is a prospective multicentre observational trial that includes women with early-stage breast cancer in the Netherlands.
⇒ The LIMA trial was designed to resemble daily clinical practice which facilitates translation and adds to generalisability of results.
⇒ The LIMA trial has a low burden for recruited patients.

## INTRODUCTION

Neoadjuvant chemotherapy (NAC) has become an important treatment strategy for early stage patients with breast cancer. Compared with adjuvant chemotherapy, NAC potentially results in less extensive surgery of both breast and axilla, without compromising distant recurrence, breast cancer survival or overall survival (OS).[1–3] The degree of response depends largely on sensitivity to therapy and is known to vary in the different breast cancer subtypes, where the highest pathological complete response (pCR) rate is reached within the human epidermal growth factor receptor 2 (HER2)-positive and the triple negative (TN) subtypes.[4–7]

With the neoadjuvant approach, the tumour is left in situ during chemotherapy, which enables evaluation of treatment efficacy. Whether pCR is achieved has an impact on patient prognosis, although prognostic

value may vary depending on pCR definition and tumour subtype.[4] However, the binary pCR measure ignores differences in prognosis within patients with residual disease. For a more comprehensive evaluation of tumour response after NAC, the residual cancer burden (RCB) was therefore developed, which has shown to be prognostic in all phenotypic subtypes of breast cancer.[8 9]

Although important for prognosis, evaluation of the response to NAC is typically only provided in the post-NAC surgical resection specimen, leaving only room for tailoring the treatment postsurgery, that is, adjuvant therapy. In the optimal situation, reliable information on tumour response is obtained during, or even before start of, NAC treatment providing the opportunity to tailor the neoadjuvant and surgical treatment to the observed tumour response.

Different methods for predicting tumour response prior to surgery are available in daily clinical practice, for example, physical examination, ultrasound, positron emission tomography/CT and dynamic contrast enhanced MRI (DCE-MRI) of the breast. The sensitivity of DCE-MRI for predicting pCR after NAC is reported to range between 65% and 91% and specificity is reported to range between 81% and 88%.[10–12] In clinical practice, these are generally not considered high enough to guide treatment decisions, as missed residual disease and inappropriate adjustment of treatment could have a detrimental effect on patient's prognosis. For instance, if a physician adopts a wait-and-see approach instead of surgery on the basis of complete tumour response at DCE-MRI, it may result in undertreatment and early relapse if residual cancer is actually still present in the breast.

A method to improve the accuracy of MRI is using various different imaging protocols in one single session (multiparametric MRI). Hence, the MRI registers information associated with various aspects of tumour biology (proliferation, angiogenesis and metabolism). By adding diffusion weighted imaging to the MRI protocol, intratumoural cellularity can be assessed as well, which may improve the value of MRI before, during and after NAC.[13 14]

However, multiparametric MRI is only able to visualise macroscopic disease. To optimise personalised response monitoring, some provision for analysis of microscopic residual disease is needed as well. Repeat core biopsies of the tumour bed during treatment has, however, proven to be hardly feasible in the clinical setting.[15]

In contrast, liquid biopsies taken from patients' blood are minimally invasive and can contain information from all parts of the tumour, thus potentially capturing intratumoural heterogeneity. Liquid biopsies are therefore considered a promising tool for prediction of treatment response.[16] Nonetheless, the technique is not yet part of standard clinical practice during NAC. Blood samples of cancer patients can contain circulating tumour cells (CTCs) and circulating DNA. The total cell-free DNA (cfDNA) can contain DNA from different sources.[17] When mutations that are associated with the malignant tumour are found in this cfDNA, this is called circulating tumour (ctDNA). Both the total cfDNA and mutations found in ctDNA can contain information on tumour load and tumour biology, which may be of importance for response prediction and prognosis. In patients with breast cancer who are treated with NAC, the presence of CTCs in their blood prior to NAC as well as prior to surgery is associated with worse disease-free survival (DFS) (HR, 2.47; 95% CI 1.95 to 3.14) and OS (HR, 2.55; 95% CI 1.91 to 3.39).[18] In a recent study in patients with triple negative breast cancer treated with NAC, who had residual disease at surgery, an increasing CTC count after surgery was correlated with inferior distant DFS (HR, 1.07; 95% CI 1.01 to 1.13), DFS (HR, 1.11; 95% CI 1.03 to 1.19), and OS (HR, 1.09; 95% CI 1.02 to 1.17).[19]

When serial blood samples are taken during treatment, the short half-life of ctDNA (less than 2 hours) allows for changes to be detected quickly and this facilitates dynamic response prediction.[20] Tracking of ctDNA mutations during neoadjuvant treatment can give information on presence and load of residual disease as well as associated risk of distant recurrence and mortality.[21] ctDNA analysis during treatment may also detect emerging resistance mechanisms, thus allowing the efficacy of anticancer treatments to be monitored.[22 23] Because driver mutations in breast cancer can be present at very low frequencies, especially in early stages of the disease, highly sensitive assays are necessary.[24] In addition to mutations, epigenetic changes are also important for cancer evolution. Methylation can also be detected in blood samples of patients with breast cancer and have additional prognostic value,[25] which may add to more accurate prediction of treatment response. Although literature on the correlation between methylation and prognosis is not as extensive as that for ctDNA and CTC's, one study did show a significantly worse OS rate at 100 months (78% vs 95%; p=0.002) for patients with breast cancer with methylated DNA detected in their blood compared with patients without.[26] Another study reported that early clearance of methylated DNA in the blood occurred in patients with breast cancer with pCR (n=4), and longer persisting methylated DNA in the blood occurred in patients with partial response (n=17).[27]

In summary, both MRI and liquid biopsies have been assessed individually confirming their potential to be used in response prediction and evaluation of neoadjuvant breast cancer treatment prior to surgery. Little is known about the combined value of these two techniques to improve prediction of response to NAC so that they can guide personalised treatment decisions. One study by Magbanua *et al*[28] found that adding ctDNA information early during treatment to the MRI predictor functional tumour volume (FTV) resulted in a numerical but not statistically significant increase in performance for pCR prediction. The additive value of ctDNA to MRI to predict response to NAC is thus not unequivocally demonstrated, and further research in this field is required. Our study may add to fine-tuning working hypotheses for follow-up

studies that may ultimately lead to practical guidelines, as its design allows for easy translation.

## METHODS

### Study objectives

The primary objective is to explore to what extent the combination of multiparametric MRI and liquid biopsies prior to, during and after completion of NAC, are able to predict RCB after NAC in addition to conventional clinical and pathological information.

Secondary objective is to use the strategy from the primary objective to predict alternative outcome measures: ypT0 ypN0 (ie, absence of invasive cancer and in situ cancer in the breast and axillary nodes), ypT0/is ypN0 (ie, absence of invasive cancer in the breast and axillary nodes, irrespective of ductal carcinoma in situ), ypT0/is (ie, absence of invasive cancer in the breast irrespective of ductal carcinoma in situ or nodal involvement) and residual lesion volume on DCE-MRI following NAC.

### Study design

This is a prospective multicentre observational study in patients with breast cancer undergoing NAC. The study has been approved by the Medical Ethics Review Committee of the University Medical Center Utrecht (19-396, NL67308.041.19). Standard Protocol Items: Recommendations for Interventional Trials guidelines were followed.[29] In the LIMA study, the complementary expertise of investigators in the MRI and liquid biopsy field have been combined into a consortium. The study participants will be recruited in four different Dutch hospitals. Potential study participants are screened by their treating physicians. Written informed consent will be obtained from all participants by their physician or research nurse. All participants will undergo NAC followed by surgery according to the Dutch oncology guidelines.[30] Study duration is from diagnosis of invasive breast cancer until the pathological assessment of the resection specimen after surgery.

### Patient and public involvement

Patients and public were not involved in study design. Results will not be directly disseminated to participating patients because of the unclear clinical relevance to their individual case. Results will be disseminated according to Findable, Accessible, Interoperable and Reusable (FAIR) data principles.

### Study population

In order to be eligible to participate in the study, a subject must meet all inclusion criteria and none of the exclusion criteria. We aim to include 100 patients.

Inclusion criteria:

Female patients aged 18 years or older.

1. Histologically proven invasive breast carcinoma.
2. Planned to receive NAC (and in case of a HER2-positive tumour: addition of trastuzumab and/or pertuzumab).

Exclusion criteria:

► Breast cancer oestrogen receptor (ER)-positive and HER2-negative by immunohistochemistry and Bloom and Richardson grade 1.
► Inflammatory breast cancer.
► Distant metastases on PET/CT.
► Prior ipsilateral breast cancer (contralateral breast cancer >5 years ago is allowed).
► Other active malignant diseases in the past 5 years (excluded squamous cell or basal cell carcinoma of the skin).
► Pregnancy or lactation.
► Contra-indications for MRI according to standard hospital guidelines.
► Contra-indications for gadolinium-based contrast-agent, including known prior allergic reaction to any contrast-agent, and renal failure, defined by a glomerular filtration rate $<30\,mL/min/1.73m^2$.

### Study procedures

An overview of the study procedures is shown in figure 1. All patients will undergo a PET/CT scan before the start of NAC to ensure no metastases are present at distant sites.

### MRI acquisition and analysis

MRI will be performed prior to, during (approximately halfway) and after NAC but before surgery. MRI will take place on 3 Tesla field strength scanners with a standardised scanning protocol. All MRI scans will be centrally revised by an experienced breast radiologist, blinded to predictors and primary outcome. Tumour imaging characteristics including BI-RADS descriptors and tumour dimensions in three directions will be recorded in the electronic case report form (eCRF). We will implement robust apparent diffusion coefficient mapping using standardisation of diffusion weighting factors (b values). Quantitative imaging features will be extracted automatically from tumour and healthy tissues (reflecting microenvironment). These methods will be developed and extended from previous studies.[31] Optionally, the impact of adding PET features and MRI conductivity features may be explored. PET features and MRI conductivity features will be explored/added if >75% of centres is able to provide these features; technical limitations and workflow considerations in hospitals may limit the availability of these additional features.

### Liquid biopsies

Blood samples will be taken from the patients before administration of every chemotherapy cycle, and after completion of NAC prior to surgery. Because the optimal time point for liquid biopsy analysis in the neoadjuvant treatment of non-metastatic breast cancer is still unknown, multiple liquid biopsies will be taken at multiple time points over the course of the treatment. This also allows for close monitoring of trends over the course of time.

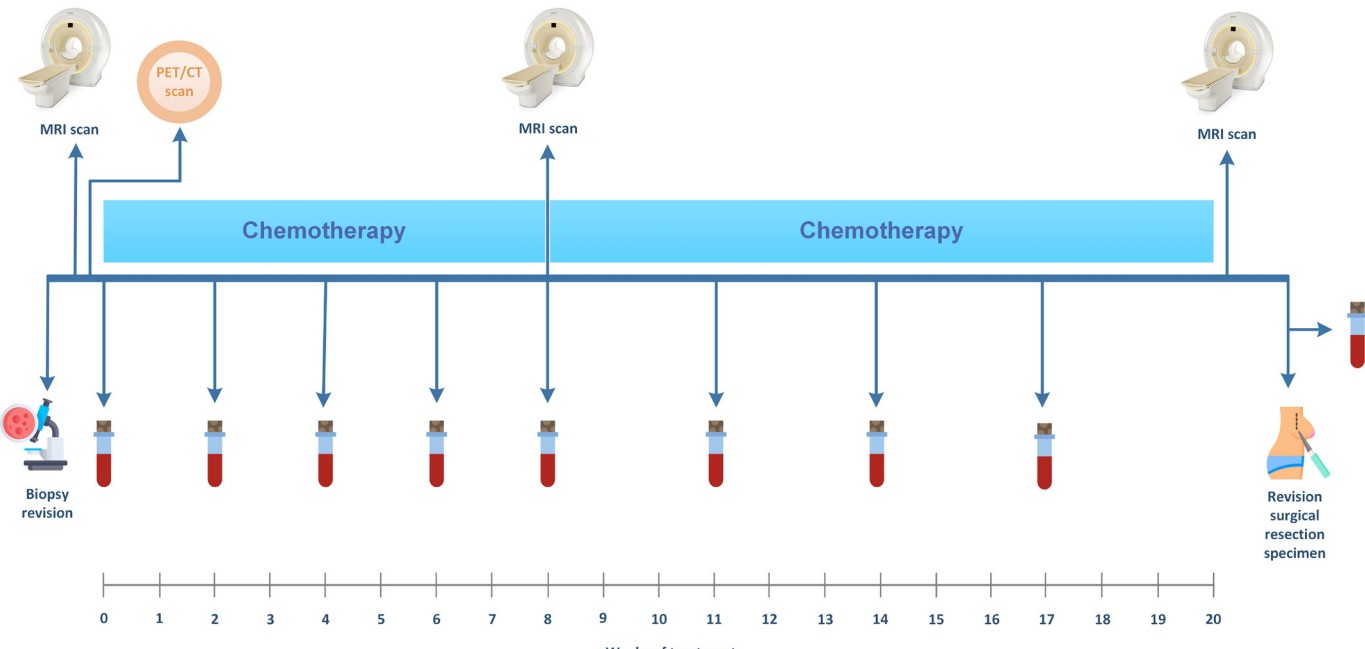

**Figure 1** Schematic overview of the study procedures. All patients undergo an MRI of the breast and a whole body positron emission tomography/CT before treatment. MRI scans are also performed during and after treatment. Blood samples are collected before every chemotherapy cycle and before surgery.

Blood samples will be drawn into blood collection tubes containing a preservation fluid. The ctDNA blood samples will be centrifuged at a central location and following a standard protocol of 10 min at 1600g. They are then stored −80°C before further processing. Liquid biopsy analyses take place in the lab of Philips in Eindhoven. After transport they are centrifuged at 16 000g. All technicians will be blinded to primary and secondary outcome measures, as well as predictors. Every sample has a unique identifier so that technicians are blinded to study participant number and longitudinal order until data collection is completed. For the analysis of the ctDNA a prespecified mutation and methylation panel will be used (online supplemental information). We will predominantly rely on a mass spectroscopy system.[32] Since mass spectroscopy is not suited to detect copy number variations, we will use digital droplet PCR (ddPCR) to detect ERBB2 amplification.[33] The ddPCR method can also be used to detect mutations that are not being picked up by the mass spectroscopy system, and this will be used for PIK3CA mutations (H1047R, E545K, E542K). CTCs will be determined at all time-points. To isolate and analyse CTCs, the blood will be filtered to reduce the amount of candidate cells by a size and compressibility filter step. After staining, the cells are scanned on a slide to identify the cells which meet the criteria to classify as CTC.[34 35]

**Pathological evaluation**

All pathology reviews will be centralised at UMC Utrecht and performed by a dedicated breast pathologist with >20 years of experience. Central review will be performed on the pre-NAC needle biopsies and the post-NAC surgical resection specimen. Blinding to results for research purposes will be performed, that is, the researchers that assess the outcome variables (pathology) do not have access to the potential candidate predictors and the other way around.

**Diagnostic biopsy**

Tumour sections will be stained by H&E staining for initial pathology diagnosis including histological type and grade according to the Nottingham modification of the Bloom and Richardson method.[36 37] Immunohistochemistry staining for tumour markers will be routinely performed on the most representative paraffin block. ER, PR and HER2 will be interpreted according to Dutch guidelines.[30] ER and PRs receptor are considered positive if >10% of nuclei stain positive. Tumours with 3+ HER2 score (strong homogeneous membrane staining in >10% of tumour cells) or HER2 gene amplification are considered HER2 positive on central revision.

**Surgical resection specimen**

Management of the resection specimens will be carried out according to the routine clinical protocol. RCB takes the dimensions of the primary tumour bed into account, as well as cellularity, percentage of in situ disease, number of positive lymph nodes and diameter of the largest lymph node metastases. These items will be reviewed in the surgical resection specimen by a trained pathologist. Calculation of the RCB will be done according to the guidelines and using the calculator provided by the MD Anderson website.[38]

## Data collection and safety reporting

Treatment regimen and patient characteristics including age, height, weight, menopausal status and stage by the American Joint Committee on Cancer[39] will be recorded in the eCRF. For the eCRF a Good Clinical Practice-compliant data capture tool will be used, which has direct input validation, edit checks and automatic saving. Personal data will be saved in an encrypted software system with two-factor authentication and limited access for designated study team members only. This study will follow the FAIR principles in handling and storage of data.[40] A data safety monitoring board is not implemented because the study is in the negligible-risk category. For this reason, only two adverse events that can be related to the study procedures will be reported as (serious) adverse events: allergic reactions to contrast agents that are administered during the MRI scans and (thrombo)phlebitis as a result of the intravenous catheter. According to regulations, a medical doctor is always present at the MRI unit when contrast is given. Study monitoring is coordinated by the sponsor and bi-annual monitoring visits are planned.

The start date of the study (first patient included) was 2 January 2020 and the expected end date is September 2022.

## Statistical analysis plan

A formal sample size and power calculation are impossible for this type of study with a large number of candidate predictor features in relation to the number patients, because meaningful (co-)variance data is lacking to feed informative simulation studies. Nevertheless, similar studies of this size have succeeded in generating clinically meaningful predictive signatures.[41] Furthermore, our primary endpoint (RCB) is continuous, increasing the effective sample size compared with a binary outcome (such as pCR). Finally, inclusion of 100 patients is also what we deem feasible based on the number of patients with breast cancer treated with NAC in our region in a 2-year time period.

The primary analysis population will include all patients who receive at least one cycle of neo-adjuvant treatment and have the primary outcome assessed (ie, residual breast cancer burden). Patterns of missing data will be inspected and if necessary we will use established methods for multiple imputation to account for missing data under the missing at random assumption.

To meet our primary objective we will estimate the over-optimism corrected mean square error and associated 95% CIs for predicting RCB in the primary analysis population using all candidate predictors from the clinical data, biopsy data and imaging data with or without the features from the liquid biopsies. These scenarios are tested at three time points: before, half way through and at the end of NAC treatment. We will use the prediction scenarios with and without liquid biopsies features to examine their additive value to the MRI-clinical-pathology-based model.

To develop the optimal and most parsimonious prediction model for each scenario, we will primarily make use of Least Absolute Shrinkage and Selection Operator penalised linear regression techniques, using bootstrapping to obtain the penalty value that minimises the mean square error in RCB prediction. This will be repeated in each imputation dataset, and the optimal models from each imputation dataset will then be averaged to obtain one final optimal model for each analysed scenario. We will repeat all these modelling steps under an additional bootstrap resampling scheme for an additional internal validation step to optimally correct for over-optimism.

Secondary to the estimation of the mean square error of the models, we will assess the models' performance in other ways as well, including: (1) agreement between predicted and actual observed RCB to assess calibration using scatterplots and linear regression analysis; (2) performance of the prediction models when the predictions of RCB as a continuous measure are compared with clinically relevant subgroups of actual RCB using receiver operating curves (discrimination) and decision curve analysis (net benefit). For our secondary objectives we will use similar data-analysis approaches.

## DISCUSSION

With the neoadjuvant approach, the tumour is left in situ during chemotherapy. The extent to which the tumour of an individual patients responds to NAC is highly variable. This variability in response means a certain NAC regimen could be overtreatment in one patient, but undertreatment in another. To define the right treatment approach for an individual patients, and to correctly balance the treatment related side effects and oncological safety, accurate prediction of response is essential. Response prediction could be used to personalise treatment for breast cancer treated with NAC in different scenarios. After completion of NAC, but before surgery, reliable tumour response evaluation is essential for facilitating de-escalation of the surgical treatment of both breast and axilla. If this evaluation is accurate enough, a wait-and-see approach may even be imaginable, sparing patients surgery-associated morbidity.

When response to NAC is assessed at earlier time points during treatment, it can provide a different set of opportunities for tailoring the treatment to individual patients' needs. An inadequate tumour response at interim evaluation may guide the treating physicians to opt for a different (non-cross resistant) chemotherapy regimen, choose a different type of systemic treatment, or adapt (the timing of) surgical intervention. Chemotherapy treatment is associated with comorbidities and reduced quality of life in patients with breast cancer.

Excellent response at interim evaluation could also be a reason for adapting (the timing of) surgical intervention or may make chemotherapy de-escalation possible, thereby sparing patients unnecessary side effects.

Especially prediction of tumour response before start of any treatment is challenging, but could have a major impact on determining the treatment strategy. Leaving the tumour in situ during NAC can carry risks in aggressive tumours that will not respond to NAC. If this (lack of) response to NAC could be reliably predicted beforehand, more effective treatment options may be adopted.

At this point, however, no method for response prediction available in clinical practice is deemed accurate enough to guide this personalised treatment approach. New strategies for predicting response to NAC include image guided tumour bed biopsy for detecting pCR in the breast after NAC in patients with partial or complete radiological response. Unfortunately, studies have shown relatively high false negative rates ranging from 17.8% to 37% for detecting pCR (defined as ypT0), which means tumour bed biopsies cannot (yet) be used to safely omit surgery after NAC. This may be explained by the fact that tissue biopsies are prone to sampling error, due to intratumoural spatial heterogeneity.[42] The invasive nature of tissue biopsies is also a drawback for clinical implementation.

Both multiparametric MRI and liquid biopsies are non-invasive methods for the evaluation of response that are valuable for the prediction of response to NAC. In the LIMA study these techniques are uniquely combined to fully exploit the complementary information they may hold.

A study by Magbanua *et al*[28] studied the combined use of ctDNA and MRI to predict pCR in patients included in the I-SPY 2 TRIAL (NCT01042379). They found an increase in area under the curve by adding ctDNA to an MRI-derived FTV model after 3 weeks of paclitaxel-based therapy, but the increase did not reach statistical significance. Functional tumour volume and ctDNA both did remain significant predictors of distant recurrence free survival in an exploratory multivariable analysis. Our study may add to these results in several aspects. We opted for a study design that is as close to clinical practice as possible and does not include regular study visits since blood is drawn from the intravenous catheter that is already in place during regular chemotherapy treatment appointments. Our patients are treated according to the most recent standard clinical guidelines. Therefore our study design reflects daily clinical practice, which will add to the generalisability of our findings.

Second, the trend that values of liquid biopsy predictors follow between different timepoints may hold important information, apart from these values themselves. Because our study has a liquid biopsy data point at every chemotherapy cycle, meaningful trends can be obtained which could lead to better predictions.

Thus, we also account for the fact that the optimal time points and intervals to assess ctDNA in the neoadjuvant setting are currently unknown.

There are a few useful things to consider in translating this study design to a clinical practice situation. Blood samples are analysed in an external lab which may come with some logistical challenges. Standardised panels will be used for ctDNA analysis. Some breast cancers may not carry any of the mutations in the panel. At this point the frequency of the methylation markers in early-stage breast cancer is unclear, and methylation markers may not be present in all patients. Therefore, a distinction between actual absence of any ctDNA versus the absence of ctDNA that can be detected by the panels, cannot be made. Additionally, specific patients are excluded: patients with B&R grade 1 hormone receptor positive breast cancer are excluded because of the poor NAC treatment results that have been reached for this subtype, and the proposed systemic treatment de-escalation prescribed in current guidelines. Patients with inflammatory breast cancer and recent other malignancies are excluded because these could lead to misinterpretation of ctDNA results. Pregnant or lactating women are excluded because their breast tissue on MRI would be influenced too much. Patients with a contra-indication for MRI or contrast are excluded for their safety.

This study is one of the first to combine multiparametric MRI with liquid biopsies to predict response to NAC in breast cancer. If the results of this study show proof-of-concept for combining these two techniques for accurate response prediction, larger follow-up studies can be designed to validate the value of these combined modalities in daily clinical practice.

## ETHICS AND DISSEMINATION

Medical Research Ethics Committee Utrecht has approved this study (NL67308.041.19). Informed consent will be obtained from each participant. All data are anonymised before publication. The findings of this study will be submitted to international peer-reviewed journals.

**Author affiliations**
[1]Image Sciences Institute, University Medical Centre Utrecht, Utrecht University, Utrecht, The Netherlands
[2]Department of Medical Oncology, University Medical Centre Utrecht, Utrecht University, Utrecht, The Netherlands
[3]Julius Center for Health Sciences and Primary Care, University Medical Centre Utrecht, Utrecht University, Utrecht, The Netherlands
[4]Department of Pathology, University Medical Centre Utrecht, Utrecht University, Utrecht, The Netherlands

**Acknowledgements** We would like to acknowledge Dr W B Veldhuis for input from a radiological standpoint and we are grateful for the support of the trial bureau of division Imaging & Oncology of the UMC Utrecht, with a special thanks to Shanta Kalaykhan. This study is funded by European Union Horizon 2020 research and innovation programme under grant agreement no. 755 333 (LIMA). Other members of this project are employees of Institut national de la santé et de la recherche médicale (INSERM), Montpellier Cancer Institute France, Angle Europe Limited,

ALS Automated Lab Solutions GmbH, AGENA Bioscience GmbH, DiaDX, Stilla Technologies, Philips Electronics Nederland B.V. and Philips GmbH.

**Contributors** KGAG and EvdW conceived the study; KGAG is the principal investigator of the grant; LMJ, EvdW, SGE and KGAG designed the final study protocol; PJvD and BBMS helped in the design of the final study protocol; LMJ and KGAG coordinated ethics approval; LMJ coordinated collaboration among investigators from all institutions; PJvD, EvdW and KGAG provided the domain knowledge expertise; MHAJ contributed to the technical design; SGE provided biostatistical and epidemiological support; LMJ, PJvD, BBMS and EvdW provided clinical input and perspectives to the qualitative aspects of the study; LMJ drafted the initial manuscript; EvdW, KGAG, and SGE revised the initial manuscript draft; all authors read and approved the final.

**Funding** Funding from the European Union Horizon 2020 research and innovation programme under grant agreement no. 755 333 (LIMA).

**Competing interests** None declared.

**Patient and public involvement** Patients and/or the public were not involved in the design, or conduct, or reporting, or dissemination plans of this research.

**Patient consent for publication** Not applicable.

**Provenance and peer review** Not commissioned; externally peer reviewed.

**ORCID iD**
Liselore M Janssen http://orcid.org/0000-0003-1157-8978

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
