## [Reviewer comments · BMJ Open]

ARTICLE DETAILS

TITLE (PROVISIONAL)	Improving prediction of response to neoadjuvant treatment in patients with breast cancer by combining liquid biopsies with multiparametric MRI: protocol of the LIMA study, a multicenter prospective observational cohort study
AUTHORS	Janssen, Liselore; Suelmann, Britt; Elias, Sjoerd; Janse, Markus; van Diest, P.J.; van der Wall, Elsken; Gilhuijs, Kenneth

VERSION 1 – REVIEW

REVIEWER	Stanhope, Edward Staffordshire University
REVIEW RETURNED	23-Feb-2022

GENERAL COMMENTS	Introduction: Page 4 Line 3-21: I commend the authors for a clear and well-written introduction to neoadjuvant chemotherapy and the importance of prognostic testing. Page 4 Line 29-31: You have indicated that 'different methods' exist in clinical practice to predict tumour response prior to surgery but only provide a single example. Could you provide several examples for those who may not be familiar with the methods used? Page 4 Line 30-32: You refer to DCE-MRI in line 30 but refer to the sensitivity and specificity of CE-MRI in lines 31-32. Consider using the same acronym for consistency/clarity. Page 4 Line 36: You discuss that missed residual disease and inappropriate adjustment of treatment can have detrimental effects. Could you identify specifically what these might be? Page 4 Line 39: 'hence, the MRI is registers...' remove 'is' Page 4 Line 31-51: I commend the authors on providing an excellent rationale for investigating methods to improve magnetic resonance imaging and for demonstrating the clinical utility of these assessments. Page 4 Line 51: Could the authors provide a reference for the validity/accuracy/use of liquid biopsies used in the way stated Page 4 Line 59-60: Can the authors be more specific about what is considered 'important information Page 5 Line 3: add patients to 'triple negative (TN) breast cancer...' Page 5 Line 6: Can the strength and precision of the correlation with DDFS, DFS and OS be provided Page 5 Line 15-16: Provide a reference for 'Because driver mutations in breast cancer can be present at very low frequencies, especially in early stages of the disease, highly sensitive assays are necessary.'
---

	Page 5 Line 18-19: What is the specific prognostic value of Methylation? Can this also be linked to/associated with DDFS, DFS and OS? Methods: Page 5 Line 37-48: The authors are commended on the clarity of their primary and secondary outcomes. Page 6 Line 18: Is there any rationale in the decision to recruit 100 participants? [Some rationale has been offered on page 8 line 22-29, but is there any further rationale for 100 specifically ie funding, resource, time constraints etc?] Page 14: Timepoints on the figure would be useful for interpretation Page 7 Line 3-4: Under what conditions would PET features and MRI conductivity features be explored/added? Can justification be provided for why this would be optional? Page 7 Line 16-19: the researchers are commended for using blinding to minimise bias Page 7 Line 42-43: Can a reference be provided for the 'Nottingham modification of the Bloom and Richardson method' Page 8 Line 5: Consider defining what AJCC TNM is/or provide reference to a paper
--	---

REVIEWER	Wilting, Saskia
REVIEW RETURNED	14-Jun-2022

GENERAL COMMENTS	In general I found the study protocol clear however I do have some remarks/questions as outlined below: Major comments: 1) More details should be included on the exact cfDNA analyses and the used panels. How many patients are expected to be ctDNA positive before start of any treatment taking into account only the markers included in the panel? This is important information as this will greatly impact the study. Absence of ctDNA could mean 2 things: 1) levels of ctDNA in the blood are undetectably low or 2) the tumour does not carry any of the markers in the panel. How will the authors deal with this, as it can be expected that prognosis/prediction will be very different in these 2 patient categories. To tackle this the authors may consider to also analyse the diagnostic biopsy with the same panels to see if a positive signal is present in the tumour to begin with. 2) Although I understand the difficulty with power calculations in these settings I think the authors could at least calculate the minimal effect size they will be able to detect by including 100 patients. In addition, coming back to my first comment, if not all tumours carry alterations covered by the panels this will also negatively impact the number of patients that can actually be included in the analysis. This also needs to be taken into consideration when imputing missing data, as the assumption that this is random does not hold when it actually reflects the absence of markers in the tumour itself (biology and not random). 3) How (number of centrifugation steps, centrifugation speed) and where (central location or per institution) will blood samples be processed to plasma? We know from other studies that differences in protocols can influence the cfDNA quality and subsequently analysis outcomes.
--

	4) Will CTCs be determined at all time points? Is there a reference describing this CTC enumeration method? Why not use a certified method like the CellSearch? Minor comments: 1) liquid biopsy can refer to a sample of any bodily fluid. Therefore I would make clear from the start that you will be investigating blood. 2) cirDNA is really not an often used abbreviation and is confusing as this could also refer to circular DNA instead of circulating DNA. I would therefore urge the authors to use the term cfDNA or ccfDNA for circulating cell-free DNA. 3) page 3, line 31 misses a D in DCE-MRI
--	--

VERSION 1 – AUTHOR RESPONSE

Reviewer: 1

Dr. Edward Stanhope, Staffordshire University Comments to the Author:

Introduction:

- Page 4 Line 3-21: I commend the authors for a clear and well-written introduction to neoadjuvant chemotherapy and the importance of prognostic testing.

We would like to thank the reviewer for the kind words on our introduction. We would also like to thank the reviewer for the constructive comments and suggestions below, which we have all taken into consideration.

- Page 4 Line 29-31: You have indicated that ‘different methods’ exist in clinical practice to predict tumour response prior to surgery but only provide a single example. Could you provide several examples for those who may not be familiar with the methods used?

We edited page 3 line 22, additions in bold:

“Different methods for predicting tumor response prior to surgery are available in daily clinical practice, e.g., physical examination, ultrasound, PET/CT and dynamic contrast enhanced magnetic resonance imaging (DCE-MRI) of the breast”

- Page 4 Line 30-32: You refer to DCE-MRI in line 30 but refer to the sensitivity and specificity of CE-MRI in lines 31-32. Consider using the same acronym for consistency/clarity.

We added to page 3 line 23 “D” to change it to: “DCE-MRI” in both sentences. The reference referring to CE-MRI was removed. It did not change the stated sensitivity and specificity values.

- Tang S, Xiang C, Yang Q. The diagnostic performance of CESM and CE-MRI in evaluating the pathological response to neoadjuvant therapy in breast cancer: a systematic review and meta-analysis. The British journal of radiology. 2020;93(1112):20200301.

- Page 4 Line 36: You discuss that missed residual disease and inappropriate adjustment of treatment can have detrimental effects. Could you identify specifically what these might be?

We added to page 3 line 27: “For instance, if a physician adopts a wait-and-see approach instead of surgery on the basis of complete tumor response at DCE MRI, it may result in undertreatment and early relapse if residual cancer is actually still present in the breast.”

- Page 4 Line 39: ‘hence, the MRI is registers...’ remove ‘is’

Done

- Page 4 Line 31-51: I commend the authors on providing an excellent rationale for investigating methods to improve magnetic resonance imaging and for demonstrating the clinical utility of these assessments.

We would like to thank the reviewer again for the kind words and are very pleased with this positive feedback.

- Page 4 Line 51: Could the authors provide a reference for the validity/accuracy/use of liquid biopsies used in the way stated

We added the reference to page 3 line 43 “Jongbloed EM, Deger T, Sleijfer S, Martens JWM, Jager A, Wilting SM. A Systematic Review of the Use of Circulating Cell-Free DNA Dynamics to Monitor Rspnse to Treatment in Metastatic Breast Cancer Patients. *Cancers*. 2021;13(8):1811.”

- Page 4 Line 59-60: Can the authors be more specific about what is considered ‘important information

We edited page 4 line 2, additions in bold:

“Both the total cfDNA and mutations found in ctDNA can contain important information on tumor load and tumor biology, which may be of importance for response prediction and prognosis.”

- Page 5 Line 3: add patients to ‘triple negative (TN) breast cancer...’

Done

- Page 5 Line 6: Can the strength and precision of the correlation with DDFS, DFS and OS be provided

We edited page 4 line 4-10, additions in bold:

“In patients with breast cancer who are treated with NAC, the presence of CTCs in their blood prior to NAC as well as prior to surgery is associated with worse disease-free survival (DFS) (HR, 2.47; 95% CI; 1.95-3.14) and OS (HR, 2.55; 95% CI 1.91-3.39) (19). In a recent study in triple negative (TN) breast cancer patients treated with NAC and who had residual disease at surgery, increasing CTC count after surgery was correlated with inferior distant disease-free survival (DDFS) (HR, 1.07; 95% CI, 1.01-1.13), DFS (HR, 1.11; 95% CI, 1.03-1.19), and OSHR, 1.09; 95% CI, 1.02-1.17) (19).”

- Page 5 Line 15-16: Provide a reference for ‘Because driver mutations in breast cancer can be present at very low frequencies, especially in early stages of the disease, highly sensitive assays are necessary.’

We added the reference to page 4 line 17:

- Ignatiadis M, Sledge GW, Jeffrey SS. Liquid biopsy enters the clinic — implementation issues and future challenges. *Nature Reviews Clinical Oncology*. 2021;18(5):297-312.

- Page 5 Line 18-19: What is the specific prognostic value of Methylation? Can this also be linked to/associated with DDFS, DFS and OS?

We added to page 4 line 21:

“Although literature on the correlation between methylation, prognosis and ctDNA is not as extensive as that for ctDNA and CTC’s, one study did show a significantly worse OS rate at 100 months (78% vs. 95%; $p = 0.002$) for breast cancer patients with methylated DNA detected in their blood compared to patients without(26). Another study reported that early clearance of methylated

DNA in the blood occurred in breast cancer patients with pCR (n=4), and longer persisting methylated DNA in the blood occurred in patients with partial response (n=17)(27).”

(26) is this reference:

Fujita N, Nakayama T, Yamamoto N, Kim SJ, Shimazu K, Shimomura A, et al. Methylated DNA and total DNA in serum detected by one-step methylation-specific PCR is predictive of poor prognosis for breast cancer patients. *Oncology*. 2012;83(5):273-82.

(27) is this reference:

Avraham A, Uhlmann R, Shperber A, Birnbaum M, Sandbank J, Sella A, et al. Serum DNA methylation for monitoring response to neoadjuvant chemotherapy in breast cancer patients. *International journal of cancer*. 2012;131(7):E1166-72.

Methods:

- Page 5 Line 37-48: The authors are commended on the clarity of their primary and secondary outcomes.

Once again, thank you.

- Page 6 Line 18: Is there any rationale in the decision to recruit 100 participants? [Some rationale has been offered on page 8 line 22-29, but is there any further rationale for 100 specifically ie funding, resource, time constraints etc?]

We added to page 7 line 44 “Finally, inclusion of 100 patients is also what we deem feasible based on the number of breast cancer patients treated with NAC in our region in a 2-year time period.”

- Page 14: Timepoints on the figure would be useful for interpretation

Done

- Page 7 Line 3-4: Under what conditions would PET features and MRI conductivity features be explored/added? Can justification be provided for why this would be optional?

We added to page 6 line 14: “PET features and MRI conductivity features will be explored/added if a sufficient number of centers is able to provide these features; Technical limitations and workflow considerations in hospitals may limit the availability of these additional features.”

Page 7 Line 16-19: the researchers are commended for using blinding to minimise bias

Thank you again for your compliments.

- Page 7 Line 42-43: Can a reference be provided for the ‘Nottingham modification of the Bloom and Richardson method’

We added two references to page 7 line 4:

- Elston CW, Ellis IO. Pathological prognostic factors in breast cancer. I. The value of histological grade in breast cancer: experience from a large study with long-term follow-up. *Histopathology*. 1991;19(5):403-10.

- Bloom HJ, Richardson WW. Histological grading and prognosis in breast cancer; a study of 1409 cases of which 359 have been followed for 15 years. *Br J Cancer*. 1957;11(3):359-77.

- Page 8 Line 5: Consider defining what AJCC TNM is/or provide reference to a paper

We added a reference to Page 7, line 21:

- Cancer AJCo. *AJCC Cancer Staging Manual*, Eight Edition. 2018. p. 622.

Reviewer: 2

Saskia Wilting

Comments to the Author:

In general I found the study protocol clear however I do have some remarks/questions as outlined below:

General remark regarding the comments:

Very good points have been raised by this reviewer which we address below point by point. We emphasize that this study is explorative in nature. The study has already received funding from the European Commission, and the protocol has already been approved by the Institutional Review Board (IRB). Choices made in the study design pursued a careful balance between strengths, limitations, available budget, maximum run time and patient burden. As a result, at this stage it is very difficult to make changes to the protocol.

Major comments:

- More details should be included on the exact cfDNA analyses and the used panels.

More details have been included on the exact cfDNA analysis and the used panels by adding the following references in page 6 line 34-35:

- Mosko MJ, Nakorchevsky AA, Flores E, Metzler H, Ehrich M, van den Boom DJ, et al. Ultrasensitive Detection of Multiplexed Somatic Mutations Using MALDI-TOF Mass Spectrometry. *J Mol Diagn.* 2016;18(1):23-31.

- Madic J, Zocevic A, Senlis V, Fradet E, Andre B, Muller S, et al. Three-color crystal digital PCR. *Biomol Detect Quantif.* 2016;10:34-46.

Page 6 line 35-38 were edited. Additions in bold:

Since mass spectroscopy is not suited to detect copy number variations, we will use digital droplet PCR (ddPCR) for this purpose to detect ERBB2 amplification(34). The ddPCR method can also be used to detect mutations that are not being picked up by the mass spectroscopy system, and this will be used for PIK3CA mutations (H1047R, E545K, E542K).

The following tables were added in the Supplementary information:

custom UltraSEEK® Breast Cancer Panel1	
Gene	Coverage (missense mutations)
PIK3CA (33.89% freq in invasive breast cancer2)	N345K, C420R, E542K, E545K, E545Q, E545A, H1047R, H1047L
TP53 (36.34% freq in invasive breast cancer2)	R175H, R213, Y220C, R248W, R248W, R248Q, R273C, R273H
AKT1 (4.52% freq in invasive breast cancer2)	E17K, L52R
ERBB2 (1.68% freq in invasive breast cancer2)	G309E, G309A, S310F, L755R, L755S, L755_T759del, D769H, D769Y, V777L, V777L, L869R
ESR1 (0.63% freq in invasive breast cancer2)	A283V, K303R, E380Q, V392I, S436P, V534E, L536R, L536Q, Y537N, Y537S, Y537C, D538G, S576L

Breast cfDNA Methylation Panel3	
Gene	Genomic location
AKR1B1	Chr7:134459123
APC	Chr5:112737754
ARHGEF7	Chr13:111115541

BRCA1	Chr17:43125416
COL6A2	Chr21:46098888
GPX7	Chr14:37592244
HIST1H3C	Chr1:52602513
MDGI	Chr17:48578124
RASGRF2	Chr1:13173414
RASSF1A	Chr5:80960894
TM6SF1	Chr3:50340798
FOXA1	Chr5:180591531
SCGB3A1	Chr15:83107646
TMEFF2	Chr2:192194694

1 Giannoudis, A. et al. Genomic profiling using the UltraSEEK panel identifies discordancy between paired primary and breast cancer brain metastases and an association with brain metastasis-free survival. *Breast Cancer Res Treat* 190, 241-253, doi:10.1007/s10549-021-06364-8 (2021).

2 Cerami, E. et al. The cBio cancer genomics portal: an open platform for exploring multidimensional cancer genomics data. *Cancer discovery* 2, 401-404, doi:10.1158/2159-8290.CD-12-0095 (2012).

3 A. Sartori, E. K., D. Irwin, S. Joosse. in *Association for Molecular Pathology 2021 Vol. 23* 1567-1649 (Elsevier, 2021).

- How many patients are expected to be ctDNA positive before start of any treatment taking into account only the markers included in the panel? This is important information as this will greatly impact the study. Absence of ctDNA could mean 2 things: 1) levels of ctDNA in the blood are undetectably low or 2) the tumour does not carry any of the markers in the panel. How will the authors deal with this, as it can be expected that prognosis/prediction will be very different in these 2 patient categories. To tackle this the authors may consider to also analyse the diagnostic biopsy with the same panels to see if a positive signal is present in the tumour to begin with.

The frequencies of occurrence for the individual mutations are added to the table in the Supplementary information.

We added to the discussion on page 9, line 40-43:

“Standardized panels will be used for ctDNA analysis. Some breast cancers may not carry any of the mutations in the panel. At this point the frequency of the methylation markers in early-stage breast cancer is unclear, and methylation markers may not be present in all patients. Therefore, a distinction between actual absence of any ctDNA vs. the absence of ctDNA that can be detected by the panels, cannot be made.”

With regard to analyzing the diagnostic biopsy tissue: we will take this suggestion into consideration, although the implementation may depend on time constraints, budget availability and IRB approval.

- Although I understand the difficulty with power calculations in these settings I think the authors could at least calculate the minimal effect size they will be able to detect by including 100 patients. In addition, coming back to my first comment, if not all tumours carry alterations covered by the panels this will also negatively impact the number of patients that can actually be included in the analysis. This also needs to be taken into consideration when imputing missing data, as the assumption that this is random does not hold when it actually reflects the absence of markers in the tumour itself (biology and not random).

The aim of our study is not to correlate a single blood biomarker to patient outcome, but to establish a multivariable prediction model from multiple patient characteristics, imaging features, and blood

biomarkers. Although it is not possible to provide a meaningful sample size calculation due to the unknown (co-)variance structure of the data that is necessary for simulation studies, we are using penalized modelling techniques to dynamically and automatically adjust the number and predictive weight of candidate predictors to the available sample size in order to prevent overfitting and thus overoptimistic results. The result of this explorative analysis will thus guide us to prioritize follow-up studies.

With regard to missing data, the absence of a biomarker does not mean that its value is missing, but rather that the value was observed to be zero. These values will not be imputed because they are not missing at random. When a blood biomarker is not found, we intend to include that information in the prediction model; The mere fact that a biomarker value was not observed, may be predictive information as well.

As stated on page 8 line 3-5: "Patterns of missing data will be inspected and if necessary we will use established methods for multiple imputation to account for missing data under the missing at random (MAR) assumption."

- How (number of centrifugation steps, centrifugation speed) and where (central location or per institution) will blood samples be processed to plasma? We know from other studies that differences in protocols can influence the cfDNA quality and subsequently analysis outcomes.

Page 6, line 26-29 was edited, additions in bold.

"Blood samples will be drawn into blood collection tubes containing a preservation fluid.

The ctDNA blood samples will be centrifuged and stored at -80 °C at a central location and following a standard protocol of 10 minutes at 1600g. They are then stored -80 °C before further processing. Liquid biopsy analyses take place in the lab of Philips in Eindhoven. After transport they are centrifuged at 16000g."

- Will CTCs be determined at all time points? Is there a reference describing this CTC enumeration method? Why not use a certified method like the CellSearch?

We added to page 6 line 38: "CTCs will be determined at all time-points."

We added two references to page 6 line 40:

- Xu L, Mao X, Imrali A, Syed F, Mutsvangwa K, Berney D, et al. Optimization and Evaluation of a Novel Size Based Circulating Tumor Cell Isolation System. PloS one. 2015;10(9):e0138032.
- Neumann MH, Schneck H, Decker Y, Schomer S, Franken A, Endris V, et al. Isolation and characterization of circulating tumor cells using a novel workflow combining the CellSearch((R)) system and the CellCelector(). Biotechnol Prog. 2017;33(1):125-32.

An innovative new technology for CTC enumeration is investigated in this study in the context of the EU research grant.

Minor comments:

- liquid biopsy can refer to a sample of any bodily fluid. Therefore I would make clear from the start that you will be investigating blood.

Page 6 line 40 was edited, additions in bold:

"In contrast, liquid biopsies taken from patients' blood are minimally invasive and blood samples can contain information from all parts of the tumor, thus potentially capturing intra-tumoral heterogeneity.

- cirDNA is really not an often used abbreviation and is confusing as this could also refer to circular DNA instead of circulating DNA. I would therefore urge the authors to use the term cfDNA or ccfDNA for circulating cell-free DNA.

Page 3 line 45-46 and page 4 line 1-2: We have replaced cirDNA with cfDNA.

- page 3, line 31 misses a D in DCE-MRI

Please see our response to the third comment from Reviewer 1.

VERSION 2 – REVIEW

REVIEWER	Stanhope, Edward Staffordshire University
REVIEW RETURNED	26-Jul-2022

GENERAL COMMENTS	The researchers comprehensively responded to each of my previous concerns. My only follow-up is to the following item: Page 7 Line 3-4: Under what conditions would PET features and MRI conductivity features be explored/added? Can justification be provided for why this would be optional? We added to page 6 line 14: “PET features and MRI conductivity features will be explored/added if a sufficient number of centers is able to provide these features; Technical limitations and workflow considerations in hospitals may limit the availability of these additional features.” What would be considered a 'sufficient' number of centers? How many centers need to be able to provide these features to warrant the addition of PET features and MRI conductivity. Clearly stating this in the protocol a priori would help with transparency and objective decision-making regarding this study feature.
---

VERSION 2 – AUTHOR RESPONSE

Reviewer: 1

Dr. Edward Stanhope, Staffordshire University

Comments to the Author:

The researchers comprehensively responded to each of my previous concerns.

My only follow-up is to the following item:

Page 7 Line 3-4: Under what conditions would PET features and MRI conductivity features be explored/added? Can justification be provided for why this would be optional?

We added to page 6 line 14: “PET features and MRI conductivity features will be explored/added if a sufficient number of centers is able to provide these features; Technical limitations and workflow considerations in hospitals may limit the availability of these additional features.”

What would be considered a 'sufficient' number of centers? How many centers need to be able to provide these features to warrant the addition of PET features and MRI conductivity. Clearly stating this in the protocol a priori would help with transparency and objective decision-making regarding this study feature.

Reviewer: 1

Competing interests of Reviewer: None

We have edited the following sentence on page 6 line 14 in the manuscript:

“PET features and MRI conductivity features will be explored/added if a sufficient number >75% of centers is able to provide these features; technical limitations and workflow considerations in hospitals may limit the availability of these additional features.”

VERSION 3 – REVIEW

REVIEWER	Stanhope, Edward Staffordshire University
REVIEW RETURNED	12-Aug-2022
GENERAL COMMENTS	Thank you for addressing all of my previous review comments.